# Original Synthesis of a Nucleolipid for Preparation of Vesicular Spherical Nucleic Acids

**DOI:** 10.3390/nano12203645

**Published:** 2022-10-18

**Authors:** Erik Dimitrov, Natalia Toncheva-Moncheva, Pavel Bakardzhiev, Aleksander Forys, Jordan Doumanov, Kirilka Mladenova, Svetla Petrova, Barbara Trzebicka, Stanislav Rangelov

**Affiliations:** 1Institute of Polymers, Bulgarian Academy of Sciences, Akad. G. Bonchev St. 103A, 1113 Sofia, Bulgaria; 2Centre of Polymer and Carbon Materials, Polish Academy of Sciences, M. Curie-Sklodowskiej 34, 41-819 Zabrze, Poland; 3Department of Biochemistry, Faculty of Biology, Sofia University St. Kliment Ohridski, Dragan Tsankov Blvd. 8, 1164 Sofia, Bulgaria

**Keywords:** Spherical nucleic acids, vesicles, liposomes, nucleolipids, oligonucleotides, thiol-ene *click* reaction, light scattering, Cryo-TEM

## Abstract

Spherical nucleic acids (SNAs)—nanostructures, consisting of a nanoparticle core densely functionalized with a shell of short oligonucleotide strands—are a rapidly emerging class of nanoparticle-based therapeutics with unique properties and specific applications as drug and nucleic acid delivery and gene regulation materials. In this contribution, we report on the preparation of hollow SNA nanoconstructs by co-assembly of an originally synthesized nucleolipid—a hybrid biomacromolecule, composed of a lipidic residue, covalently linked to a DNA oligonucleotide strand—with other lipids. The nucleolipid was synthesized via a *click* chemistry approach employing initiator-free, UV light-induced thiol-ene coupling of appropriately functionalized intermediates, performed in mild conditions using a custom-made UV light-emitting device. The SNA nanoconstructs were of a vesicular structure consisting of a self-closed bilayer membrane in which the nucleolipid was intercalated via its lipid–mimetic residue. They were in the lower nanometer size range, moderately negatively charged, and were found to carry thousands of oligonucleotide strands per particle, corresponding to a grafting density comparable to that of other SNA structures. The surface density of the strands on the bilayer implied that they adopted an unextended conformation. We demonstrated that preformed vesicular structures could be successfully loaded with either hydrophilic or hydrophobic dyes.

## 1. Introduction

Spherical nucleic acids (SNAs) are fascinating 3D nanostructures consisting of a nanoparticle core and an oligonucleotide shell [1,2,3,4,5]. These polyvalent architectures, which can be tuned through the type, number, and orientation of the oligonucleotide strands in the shell and through a broad range of inorganic and organic cores, have quickly gained an important role in nucleic acid-containing formulations [6,7,8]. They exhibit properties that are distinctly different from those of their constituent counterparts: SNA structures can penetrate the membranes of the cells without using transfection agents, avoid many biological barriers as well as the attack by the human immune system, and deliver the cargo. They exhibit extraordinary stability in physiological environments and resistance to nuclease degradation [9,10]. This makes them effective as high-sensitivity detection systems, and also, ideal for intracellular diagnostic and therapeutic agents [11,12,13,14,15].

To successfully shuttle nucleic acids, loaded drugs, and/or diagnostic agents into cells, SNAs can be properly designed in terms of architecture, functionality, molar mass, size, hydrophilic/lipophilic balance, etc., [16,17,18]. For instance, the selection of the right nanoparticle core plays a key role and directly determines the shape, size, and biological profiles of the core–shell assemblies [19,20,21]. Strategies to design SNAs with organic cores, formed by polymers, lipids, small molecules, and proteins, have recently been developed to obtain biocompatible and biodegradable SNAs [22,23,24,25]. Among them, the liposomal SNAs, with their hollow interior and bilayer structural organization emerged as a new class of nucleic acids. They are synthesized by anchoring/intercalation of hydrophobically modified oligonucleotide strands via their hydrophobic residue into the bilayer membrane of phospholipid liposomes [23,26,27]. The advances in the understanding of liposomal SNA constructs reveal that they offer superior antitumor efficacy, higher downstream cellular activity, increased cellular uptake, faster DNA release, and higher DNA-loading capacity over the prototypical SNAs [26,28].

The liposomal SNAs define a new class of SNAs that is held together via non-covalent bonds. They are typically made from phospholipids and cholesterol (both components of the cell membranes) and a small quantity of nucleolipid. Whereas the former two are part of FDA-approved pharmaceuticals, the latter can be specially designed to serve an integral role in the function and behavior of the liposomal SNAs. However, the barrier to the therapeutic use of new materials is high, particularly for such that may have problems with clearance or unknown biodistribution. In this regard, utilization of synthetic routes to circumvent any possible contaminations in the resulting nucleolipids, which can pose a considerable challenge in their efficient removal, is of significant importance. The *click* chemistry approaches offer such a possibility. These coupling reactions are straightforward and typically proceed in mild conditions, give quantitative yields, and allow simple isolation of products. The strategy is based on separate preparations of “clickable” entities (here, an oligonucleotide strand and a lipophilic moiety) and their coupling into a novel architecture. Importantly, it makes it possible to achieve combinations of species that are inaccessible by conventional methods, e.g., phosphoramidite chemistry for the synthesis of nucleolipids. Among the various *click* reactions, the most generally used ones require metal catalysts or radical initiators, which may pose some drawbacks associated with the use of, e.g., Cu(I) in biological systems [29] or DNA degradation [30]. Herein, we employ an original synthetic approach for the preparation of a nucleolipid by covalently joining a lipid–mimetic double chain residue with a 21-base-long nucleic acid strand using initiator-free, UV light-induced thiol-ene *click* reaction. Further, by simple intercalation of the novel nucleolipid in the phospholipid bilayer of liposomes, well-defined liposomal SNA structures are obtained and fully characterized. The loading capability of the structures with respect to hydrophilic and hydrophobic substances is examined as well.

## 2. Materials and Methods

1-Hexadecanol, hexadecyl glycidyl ether, allyl glycidyl ether, SnCl_4_, potassium (cubes in mineral oil), 2-aminoethanethiol hydrochloride, poly(ethylene glycol) methyl ether thiol (PEGMET, M_w_ = 2000 g/mol), 1,2-dipalmitoyl-rac-glycero-3-phosphocholine (DPPC), and cholesterol (Chol), all products of Sigma-Aldrich, were used as received. Hexane (Sigma-Aldrich, Saint Louis, MO, USA), 1,4-dioxane (Sigma-Aldrich), methylene chloride (Fisher Scientific, Waltham, MA, USA), toluene (Fisher Scientific), benzene (Merck, Darmstadt, Germany), methanol (Sigma-Aldrich), and tetrahydrofuran (Fisher Scientific) were dried with calcium hydride and freshly distilled before use. N,N-Dimethylformamide, (DMF, ACS reagent, ≥99.8%) and dimethyl sulfoxide (DMSO, ACS reagent, ≥99.9%) were dried by molecular sieves. Deionized water was obtained by a Millipore MilliQ system and was additionally filtered through a 220 nm PTFE filter and a 20 nm cellulose filter. Thiol-functionalized oligonucleotide (ThiolC_6_-oligonucleotide) was purchased from Biomers.net GmbH. The sequence and composition as well as characterization data, according to the producer, are presented in Appendix A. Details for the synthesis and characterization of dihexadecyloxy-propane-2-ol (DHP) and spacer-modified DHP-(AGE)_n_ are given in Appendix A.

*Synthesis of the nucleolipid DHP-(AGE)_3_-oligonucleotide.* A total of 491.3 µg (61.4 nmol, 1 eq.) of thiolated oligonucleotide, and 197.4 µg (223 nmol, 3.6 eq.) of DHP-(AGE)_3_ were dissolved in a 2.5 mL solvent mixture of DMF:DMSO:H_2_O (*v*/*v* 0.45:0.45:0.1) and placed in a round-bottom flask under vigorous stirring, flushed with argon, and wrapped in aluminum foil. The UV light-induced thiol-ene *click* reaction was carried out for 1 h at 30 °C and the full (4 W) capacity of the UV irradiation device. Detailed information for the device is given in the Appendix A. The reaction mixture was cooled to room temperature and purified by intense dialysis against a mixture of DMSO:DMF (*v*/*v* 1:1) for 2 days using an 8 kDa MWCO membrane. Then the organic dialysis solvent was replaced with ultra-pure MilliQ water followed by ultrafiltration for additional 3 days yielding a clear to slightly opalescent aqueous dispersion of the nucleolipid DHP-(AGE)_3_-oligonucleotide. Finally, the sample was freeze dried. Yield: 470 µg, 85%.

*Preparation of vesicular SNAs.* Stock solutions of known concentrations of DPPC and Chol in chloroform and the nucleolipid DHP-(AGE)_3_-oligonucleotide in methanol were prepared. Predefined amounts of these solutions were placed into glass tubes to achieve 2 mM total lipid concentration, 2:1 DPPC:Chol M ratio, and 2 mol % of nucleolipid as follows: 0.04 µmol (301.90 µg) of nucleolipid, 1.30 µmol (954.2 µg) of DPPC, and 0.65 µmol (251.55 µg) of Chol. The solvent ratio was kept in the 1:9–1:10 range with respect to methanol in order to avoid precipitation. After mixing, the solvents were removed under a gentle stream of argon leaving a thin film. All traces of solvent were removed under a vacuum overnight at room temperature. The dry, thin lipid film was hydrated with 1 mL of MilliQ water and the resulting dispersion was subjected to ten freeze–thaw cycles and then extruded 15 times through polycarbonate filters (100 nm pore size) using a handle-type extruder (Avanti Polar Lipids, Alabaster, AL, USA).

*Nuclear Magnetic Resonance (^1^H-NMR).*^1^H-NMR measurements were conducted on a Bruker Avance II spectrometer operating at 600 MHz using CDCl_3_, DMSO-d6, or benzene-d₆ at 25 °C.

*Size Exclusion Chromatography (SEC)*. SEC analyses were performed on a Shimadzu Nexera HPLC chromatograph equipped with a degasser, a pump, an autosampler, an RI detector, and three columns: 10 μm PL gel mixed-B and 5 μm PL gel 500 Å and 50 Å. Tetrahydrofuran was used as the eluent at a flow rate of 1.0 mL·min^−1^ and temperature of 40 °C. The sample concentration was 1 mg·mL^−1^, and SEC was calibrated with polystyrene standards.

*Light Scattering.* Dynamic light scattering (DLS) measurements were performed on a Brookhaven BI-200 goniometer with vertically polarized incident light at a wavelength λ = 633 nm supplied by a He–Ne laser operating at 35 mW and equipped with a Brookhaven BI-9000 AT digital autocorrelator. Measurements were made at an angle of 90° and 37 °C. The autocorrelation functions were analyzed using the constrained regularized algorithm CONTIN [31] to obtain the distributions of the relaxation rates (Γ_90_). The latter provided distributions of the apparent diffusion coefficient (D_90_ = Γ_90_/q^2^) where q is the magnitude of the scattering vector given by q = (4πn/λ)sin (θ/2), n is the refractive index of the medium, and θ = 90°. The mean hydrodynamic radius was obtained by the Stokes–Einstein equation (Equation (1)):R_h_ = kT/(6πηD_90_) (1)
where k is the Boltzmann constant, and η is the solvent viscosity at temperature T in Kelvin.

*Electrophoretic Light Scattering:* The electrophoretic light scattering measurements were carried out on a 90Plus PALS instrument (Brookhaven Instruments Corporation, Hosttville, NY, USA), equipped with a 35 mW red diode laser (λ = 640 nm) at a scattering angle (θ) of 15°. ζ potentials were calculated from the obtained electrophoretic mobility at 37 °C by using the Smoluchowski Equation (2):ζ = 4πηυ/ε (2)
where η is the solvent viscosity, υ is the electrophoretic mobility, and ε is the dielectric constant of the solvent.

*Cryogenic Transmission Electron Microscopy (Cryo-TEM).* Cryo-TEM images were obtained using a Tecnai F20 X TWIN microscope (FEI Company, Hillsboro, OR, USA) equipped with a field-emission gun, operating at an acceleration voltage of 200 kV. Images were recorded on the Gatan Rio 16 CMOS 4k camera an Eagle 4k HS camera (Gatan Inc., Pleasanton, CA, USA) and processed with Gatan Microscopy Suite (GMS) software (Gatan Inc., Pleasanton, CA, USA). Specimen preparation was done by the vitrification of the aqueous dispersions on grids with a holey carbon film (Quantifoil R 2/2; Quantifoil Micro Tools GmbH, Großlöbichau, Germany). Prior to use, the grids were activated for 15 s in oxygen plasma using a Femto plasma cleaner (Diener Electronic, Ebhausen, Germany). Cryo-samples were prepared by applying a droplet (3 μL) of the dispersion to the grid, blotting with filter paper and immediate freezing in liquid ethane using a fully automated blotting device Vitrobot Mark IV (Thermo Fisher Scientific, Waltham, MA, USA). After preparation, the vitrified specimens were kept under liquid nitrogen until they were inserted into a cryo-TEM holder Gatan 626 (Gatan Inc., Pleasanton, CA, USA) and analyzed at −178 °C.

*Gel electrophoresis.* The completeness of the *click* coupling reaction was confirmed by agarose gel electrophoresis. Gels containing 1% agarose (*w*/*w*) were run on an FBSB-710 electrophoresis unit (Fisher Biotech) in 1 × Tris-Borate EDTA (TBE) buffer at room temperature and 50 V. Imaging was carried out by ethidium bromide staining and UV illumination (302 nm). Quick-Load^®^ Purple 1 kb DNA Ladder of BioLabs was used as a marker. Total amounts of 0.6 µg were loaded. The gels were imaged using a gel reader Alpha Innotech.

*Fluorescence microscopy.* The vesicular SNAs were incubated with Laurdan (Sigma Aldrich) in a final concentration of 25 μM Laurdan in DMSO for 30 min at 37 °C [32]. For the covalent binding of fluorescein isothiocyanate (FITC, final concentration 5 mM) (Sigma Aldrich) to vesicular SNAs, incubation was performed for 2 h at room temperature [33]. After incubation, vesicular SNAs were dialyzed against phosphate buffer pH 7.4 for 2 h at room temperature. Vesicular SNAs were visualized with a fluorescence microscope (GE Delta Vision Ultra Microscopy System) with a 60× immersion objective.

## 3. Results and Discussion

### 3.1. Synthesis and Characterization of the Nucleolipid

We designed a novel nucleolipid, comprising a lipid–mimetic residue covalently attached to a short (21 bases long) single-stranded DNA oligonucleotide. The synthesis was realized by an initiator-free, UV-induced thiol-ene *click* reaction using a custom-made irradiation device (see Appendix A for the description of the device). The synthetic route for the preparation of the nucleolipid is shown in Figure 1. Briefly, the lipid–mimetic anchor, 1,3-dihexadecyloxy-propane-2-ol (DHP), prepared as described elsewhere [34,35], was modified by introducing a short spacer with an average of three allyl glycidyl ether units, (AGE)_3_. The introduction of a spacer has been shown critical for further modification of the secondary hydroxyl group of DHP [27]. Furthermore, the spacer segment pushes the oligonucleotide strand away from the lipid–mimetic anchor, which gives more freedom to the former for various interactions. We selected a polyether spacer, which, in contrast to other common polyether spacers such as poly(ethylene glycol) or poly(propylene glycol), is not chemically inert: it contains allyl functionality, which can be exploited for various orthogonal modifications including radical addition of thiols. The target degree of polymerization was just a few units because poly(allyl glycidyl ether) is a hydrophobic polymer [36] and longer chains would introduce significant hydrophobicity in the final nucleolipid. The formation of the spacer was achieved by anionic ring-opening polymerization of AGE initiated by partially deprotonated DHP, thus, forming DHP-(AGE)_n_ intermediate (Figure 1a). The SEC analysis showed a monomodal molar mass distribution (M_w_^SEC^ = 1350 g.mol^−1^, M_w_/M_n_ = 1.1, see Appendix A), which is typical for the polymerization technique used. Since the molar mass from SEC is relative to polystyrene standards, the average degree of polymerization of the (AGE)_n_ spacer was determined from the ^1^H NMR spectrum of the product shown in Figure 2a. The average number of AGE units (n = 3) was calculated from the integral ratio of the signal for the methyl protons (protons *a* at 0.85 ppm) to methine protons of the allyl groups (protons *i* at 5.80 ppm).

The conjugation of the oligonucleotide to DHP-(AGE)_3_ was performed via the thiol-ene coupling reaction. The reaction is orthogonal [37], tolerant to a variety of functional groups and can be performed in aqueous media. Normally, the thiol-ene coupling reaction can be initiated by UV light, which induces the formation of a thiol radical that reacts with the carbon–carbon double bond to produce a thioether. In that aspect, the (AGE)_n_ spacer offers another benefit: since each unit of the spacer carries a double bond, the number of sites on which the coupling reaction takes place is increased. As UV light and free radicals can be harmful to nucleic acids, we performed model reactions to find optimal conditions for the thiol-ene coupling reaction (see Appendix A). Firstly, we determined that the safe UV irradiation time for the oligonucleotides is 1 h at 30 °C, λ = 365 nm, and the full capacity (4 W) of the device. Next, model reactions of DHP-(AGE)_n_ with 2-aminoethanethiol hydrochloride and poly(ethylene glycol) methyl ether thiol (PEGMET, M_w_ = 2000 g/mol), representing low- and high molar mass thiolated compounds, respectively, were performed (see Appendix A). The results revealed that 2-aminoethanthiol hydrochloride reacted with two, whereas PEGMET with only one, presumably the outermost, of the double bonds of the (AGE)_n_, spacer (see Appendix A).

The coupling reaction between DHP-(AGE)_n_ and thiolated oligonucleotide was performed at the above conditions in a large excess of DHP-(AGE)_n_ in a mixed DMSO/DMF/water solvent (Figure 1b). After the reaction was accomplished, the system was cooled down to room temperature and subjected to dialysis against DMF:DMSO to remove the excess of DHP-(AGE)_n_. Afterwards, the mixed solvent was displaced by ultrafiltration, which yielded a dilute aqueous dispersion of the resulting nucleolipid. The latter was characterized by gel electrophoresis (Figure 2b). Ethidium bromide was used for visualizing and identifying the nucleic acid bands. It has been shown to bind to single-stranded nucleic acids although less strongly than to double-stranded ones [38]. Therefore, the fluorescence was lower and the bands of the oligonucleotide and nucleolipid were weaker compared to those of the molecular weight marker. Nevertheless, the retardation of the electrophoretic mobility of the nucleolipid compared to that of the thiolated oligonucleotide, used as control, is clearly visible in Figure 2b and is consistent with the enlargement of the molecular weight of the nucleolipid.

### 3.2. Preparation of Vesicular SNAs

Vesicular SNAs have been synthesized by surface functionalization of pre-formed phospholipid liposomes with DNA strands modified with a hydrophobic tail [26] and by co-assembly of a phospholipid (1,2-dipalmitoyl-rac-glycero-3-phosphocholine, DPPC) and an original nucleolipid composed of a lipid–mimetic residue covalently attached to a DNA oligonucleotide strand [27]. Following a common route for the preparation of small unilamellar vesicles involving thin film hydration, freeze–thawing of the dispersions, and multiple extrusion through small pore size filters, we prepared vesicular structures consisting of a self-closed oligonucleotide-grafted bilayer membrane (Figure 3). In a parallel experiment plain DPPC/Chol liposomes were prepared following the same preparation procedure.

The successful incorporation of the nucleolipid was evidenced by the appearance of absorption at 260 nm, which is characteristic of nucleic acids [39], in the UV-Vis spectrum of the purified dispersion (Figure 4a). Dynamic and electrophoretic light scattering measurements further evidenced the formation of vesicular SNAs. Compared to the plain DPPC/Chol liposomes, the vesicular SNAs were smaller in size and more negatively charged (Figure 4b and Table 1). Whereas the latter can be anticipated, the size decrease upon the incorporation of the nucleolipid in the bilayer might seem somewhat counterintuitive but can be rationalized in terms of repulsive interactions between the oligonucleotide strands that create curving of the bilayer. More curved bilayers (that is, vesicles of a smaller size) would give bigger relaxation of the repulsive interactions [40]. The particles were visualized by cryo-TEM (Figure 4c). Clearly, all objects were unilamellar vesicles of spherical morphology and dimensions that were compatible with the size determined by dynamic light scattering. The bilayer membranes were intact with a constant thickness of about 5 nm, which is consistent with DPPC systems [41]. Changes in the characterization parameters of the vesicular SNAs were not detected for ca. two months, which implied enhanced colloidal stability.

The synthesized vesicular SNAs had an average number of 3880 ± 194 strands per particle (Table 1) as calculated from the feed/stoichiometry (DPPC and oligonucleotide contents) and size of the vesicles (see Appendix A). The value is larger than that of reference samples (micellar and liposomal SNAs, gold nanoparticles SNAs) [22,42,43,44,45,46] but is consistent with the relatively larger size of the present vesicular structures. The surface coverage, σ, expressed as number of strands per squared nm, σ = 0.047 nm^−2^, and converted into amount of nucleolipid per squared cm, σ = 7.8 pmol·cm^−2^ (Table 1, see Appendix A), was of the same magnitude to those obtained for previously reported SNA structures [17,22,42,43,44,45,46] as well as oligonucleotides immobilized on solid substrates [47] considering that σ is dependent on particle size and shape [1] and can be controlled by the salt [48] and nucleolipid contents.

The conformation of the oligonucleotide strands can be estimated by comparing the values of the grafting density with the critical grafting density (σ_cr_) at the transition to strongly extended, brush conformation. Regarding the oligonucleotide strands as surface-anchored random coils, the transition from the unextended to the brush conformation, which defines two concentration regimes [49,50], can be determined by σ_cr_ = 1/R_F_^2^ where R_F_ is the Flory radius (see Appendix A for the calculation of R_F_ and σ_cr_). Since for the investigated system σ (=0.047 nm^−2^) is smaller than σ_cr_ (=0.156 nm^−2^), one may conclude that the oligonucleotide strands adopt an unextended conformation and hardly interact laterally. However, as discussed elsewhere [17], none of the SNA constructs hitherto reported exhibit grafting density values larger than σ_cr_, indicating that the oligonucleotide strands have not entered the brush regime. Notwithstanding, these structures have been shown to manifest the typical SNA properties [26].

The vesicular SNAs were loaded with model compounds that exhibit diametrically opposite solubility and different properties. Fluorescein isothiocyanate (FITC), which has the potential for covalent binding with DNA [33], and laurdan—used to investigate membrane quality of phospholipid bilayers [51]—were selected as hydrophilic and lipophilic, respectively, probes. Preformed vesicular SNAs were incubated with the two dyes and, after removing the unloaded material by dialysis, were analyzed by dynamic and electrophoretic light scattering and visualized by fluorescence microscopy (Figure 5). Noteworthy, their size and ζ potential practically did not change upon loading—the values of the hydrodynamic radius and ζ potential were within the standard deviations (see Table 1). Since the vesicles were not sufficiently large (the resolution of conventional fluorescence microscopy is limited by diffraction to about 200 nm in the focal plane and to about 500 nm along the optic axis), the method provided only qualitative evidence for successful loading. The images showed bright, about micron-size, mostly spherical objects. The sizes were larger than what was observed by dynamic light scattering (see above) which can be attributed to the clustering of SNAs during the preparation step for observation. The findings revealed the potential of the vesicular SNAs to incorporate both hydrophilic and hydrophobic compounds individually and, most probably, simultaneously, in accordance with their specific structure.

## 4. Conclusions

A novel nucleolipid consisting of a lipid–mimetic double-chain residue covalently attached to a single-stranded DNA oligonucleotide was synthesized by an initiator-free, UV-induced thiol-ene *click* reaction. The reaction was performed in mild conditions using a custom-made UV irradiation device. Optimal synthetic parameters of the reaction in terms of reaction conditions, on one hand, and safety (with regard to oligonucleotides), effectiveness, and yield, on the other, were established. The introduction of a spacer bearing multiple allyl functionalities to increase the reaction sites on which the coupling reaction takes place, was of significant importance to effectively perform the coupling reaction and obtain the product in good yield. The reaction is applicable to thiolated oligonucleotides independently of the type (DNA or RNA) and sequence. Unilamellar vesicles consisting of a self-closed oligonucleotide-grafted bilayer membrane were successfully prepared by co-assembly of the nucleolipid with readily available lipids DPPC and cholesterol. The vesicles were found to carry thousands (3880 ± 194) of oligonucleotide strands per particle, corresponding to grafting density comparable to that of other SNA structures. The value of the grafting density implied that the oligonucleotide strands did not adopt a fully extended conformation. Preformed vesicles were easily and spontaneously loaded with model compounds exhibiting different solubility properties. Consequently, these structures show promise as carriers for two or more payloads—the nucleic acid component that can be designed with a specific sequence for, e.g., gene silencing, and hydrophilic or hydrophobic drugs separately or together. Such constructs can bypass the need for a co-carrier system in combination therapy and allow one to access simultaneously a gene target and a drug target.

## Figures and Tables

**Figure 1 nanomaterials-12-03645-f001:**
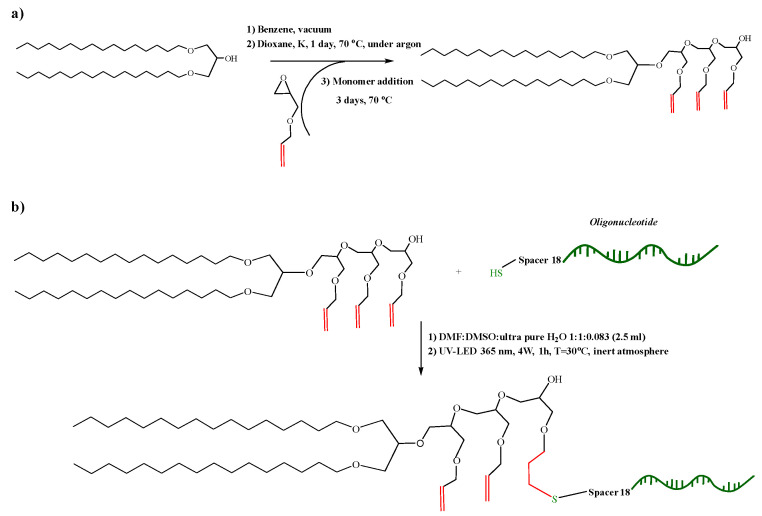
Synthetic schemes for preparation of (**a**) the DHP-(AGE)_n_ intermediate (n = 3) and (**b**) the nucleolipid DHP-(AGE)_3_-oligonucleotide.

**Figure 2 nanomaterials-12-03645-f002:**
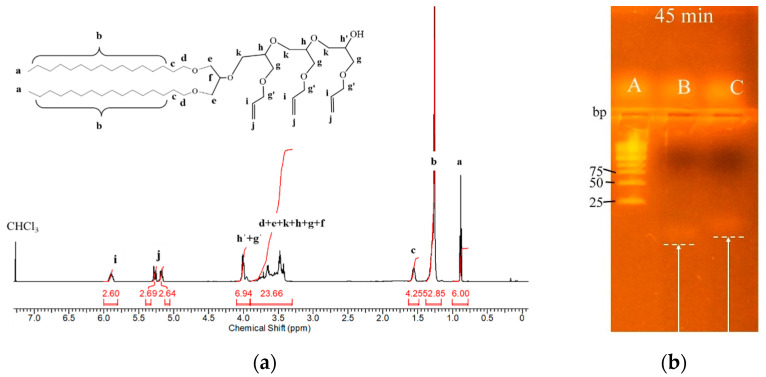
(**a**) ^1^H NMR spectrum of the intermediate DHP-(AGE)_n_ (n = 3) in CDCl_3_. (**b**) Agarose gel retardation patterns of thiolated oligonucleotide and the nucleolipid DHP-(AGE)_3_-oligonucleotide. The white arrows indicate the different retardations of the thiolated oligonucleotide (lane B) and nucleolipid (lane C). Nucleic acids were stained with ethidium bromide. Overall duration of the experiment 45 min as indicated. Lane A: molecular weight marker.

**Figure 3 nanomaterials-12-03645-f003:**
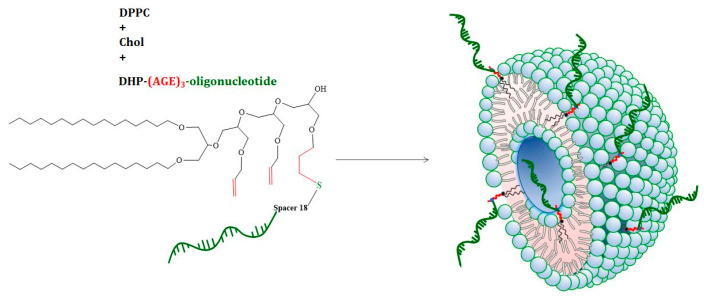
Schematics of the formation of vesicular SNAs from DPPC, cholesterol, and nucleolipid.

**Figure 4 nanomaterials-12-03645-f004:**
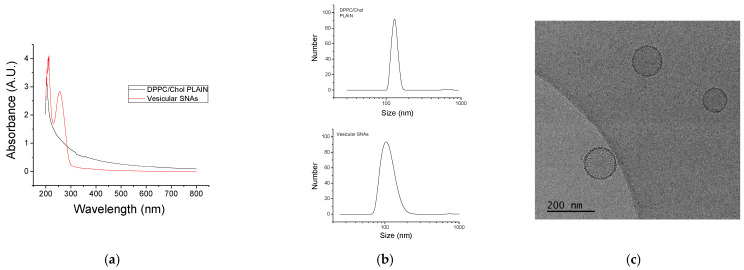
(**a**) UV-Vis spectra of vesicular SNAs and plain DPPC/Chol liposomes. (**b**) Size distributions from DLS of plain DPPC/Chol liposomes and vesicular SNAs as indicated. (**c**) Cryo-TEM images of vesicular SNAs.

**Figure 5 nanomaterials-12-03645-f005:**
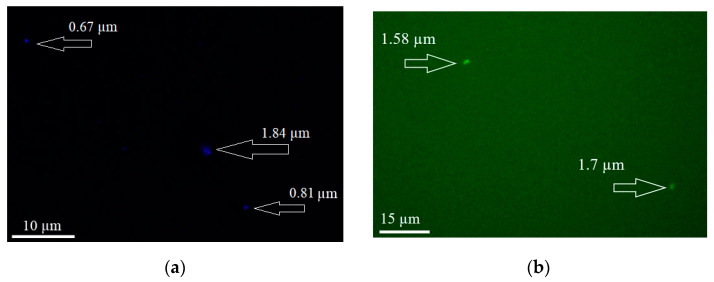
Fluorescent labeling of vesicular SNAs with (**a**) laurdan and (**b**) FITC.

**Table 1 nanomaterials-12-03645-t001:** Light scattering parameters and other characterization data of plain DPPC/Chol liposomes and vesicular SNAs, prepared by co-assembly of DPPC, cholesterol, and nucleolipid. Measurements were performed at 37 °C.

	R_h_ * (nm)	ζ Potential (mV)	Number of Strands per Vesicle	Surface Coverage	
				(nm^−2^)	pmol·cm^−2^
Plain DPPC/Chol liposomes				n.a.	n.a.
	67.4 ± 3.4	–0.34 ± 0.01	n.a		
	(0.087)				
Vesicular SNAs	57.1 ± 2.9	–9.11 ± 0.36	3880 ± 194	0.047	7.8
	(0.127)				

* PDI values in parentheses.

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
