# Peer review of "Original Synthesis of a Nucleolipid for Preparation of Vesicular Spherical Nucleic Acids"

_nanomaterials, 2022, doi:10.3390/nano12203645_

Round 1

Reviewer 1 Report

In this manuscript, the authors synthesized a nucleolipid using a UV-induced thiol-ene coupling reaction. As-prepared nucleolipid was applied to prepare a spherical nucleic acid (SNA). The manuscript can be interesting to authors in the field of drug delivery. However, the following issues need to be addressed before the acceptance of the manuscript.

1. In Line 187, the authors mentioned that on average there are 3 units. Amore detailed characterization should be performed to further understand the DHP-(AGE)n. For example, what is the distribution of the molecules with different units? This might be characterized by LC-MS or other methods.

2. Similarly, the authors should provide a more detailed characterization of the final product DHP-(AGE)n-oligonucleotide. Evidence showing the number and position of oligos on the polymer should be given. 

3. It seems that the authors only used one concentration of the nucleolipid to prepare the SNAs. Is this the optimized result? It might be interesting to change the percentage of the nucleolipid and see the impact on particle size and/or charge.

4.  In Figure 5, the size and zeta potential of the particles after dye loading should be measured.

Author Response

Reviewer 1

In this manuscript, the authors synthesized a nucleolipid using a UV-induced thiol-ene coupling reaction. As-prepared nucleolipid was applied to prepare a spherical nucleic acid (SNA). The manuscript can be interesting to authors in the field of drug delivery. However, the following issues need to be addressed before the acceptance of the manuscript.

  1. In Line 187, the authors mentioned that on average there are 3 units. Amore detailed characterization should be performed to further understand the DHP-(AGE)n. For example, what is the distribution of the molecules with different units? This might be characterized by LC-MS or other methods.

Response: A more detailed characterization is provided. As DHP-(AGE)3 can be considered in many aspects as a polymer, size exclusion chromatography (SEC) was performed to determine the molar mass distribution. An additional figure (Figure S4), description of SEC and discussion of the results are given in the Supplementary Materials as well as in lines 129-134 and 205-211. The results show a relatively narrow molar mass distribution, given by the Mw/Mn ratio of 1.1, which is typical for the polymerization technique used. Since the molar mass from SEC is relative to polystyrene standards, the average degree of polymerization of the (AGE)n spacer was determined from the 1H NMR spectrum of the product as shown in Figure 2a.

  1. Similarly, the authors should provide a more detailed characterization of the final product DHP-(AGE)n-oligonucleotide. Evidence showing the number and position of oligos on the polymer should be given.

Response: We would like to point the reviewer’s attention that the conjugation reaction was performed in large excess of DHP-(AGE)n (see lines 102-103, 241-242) so that it is very unlikely to have more than one oligonucleotide strand attached to the spacer. Furthermore, the gel electrophoresis, which is the most appropriate and widely used method to separate mixtures of DNA, RNA or proteins, showed only one band, that is only slightly retarded compared to the control (the starting thiolated oligonucleotide), which is consistent with the enlargement of the molecular weight of the resulting nucleolipid (see Figure 2b and text on lines 247-254). If there were more oligonucleotide strands attached to the spacer, either more bands, broadening of the band or stronger retardation would be observable. With regard to the position of the oligonucleotide strand, due to steric considerations and results from the model reaction with PEGMET (see Supplementary Materials), it can be anticipated that the conjugation reaction takes place at the outermost double bond of the (AGE)n spacer. The text on lines 234-240 was altered to discuss this particular nucleolipid characteristics.

  1. It seems that the authors only used one concentration of the nucleolipid to prepare the SNAs. Is this the optimized result? It might be interesting to change the percentage of the nucleolipid and see the impact on particle size and/or charge.

Response: The oligonucleotide to total lipid ratio used in our research is based on published protocols for nucleic acid containing liposomal compositions. A topic for future research could be to follow the impact of the nucleolipid content on the particle size, charge, structure and morphology. It should be noted here that the increase of nucleolipid content is not unlimited since the higher contents tend to increase the bilayer curvature and, eventually, to transition from bilayers (that is, vesicular structure) to micellar phase (mixed DPPC-nucleolipid micelles). Undoubtedly, the increase of nucleolipid content within the above range would bring about to increasing in the number of strands per particle and surface coverage as well as transition to more extended (brush) conformation of the oligonucleotide strands as discussed on lines 295-301 and 302-313.

  1. In Figure 5, the size and zeta potential of the particles after dye loading should be measured.

Response: The size and zeta potential of the vesicles after dye loading practically did not change – the values were within the standard deviations (see Table 1). The text in lines 318-322 was changed to acknowledge this finding.

Reviewer 2 Report

Using a typical click chemistry thiol-ene reaction, the author synthesized the nucleolipid DHP-(AGE)3-oligonucleotide, formulated with phospholipid DPPC and cholesterol as SNAs for model dyes encapsulation. The synthesis protocol is well-planed, with the produced compound/nanoparticle systematically characterized and identified via various techniques, including NMR, HPLC, DLS, UV-Vis, Cryo-TEM, etc., potentially bringing further applications of the SNAs such as drug or gene delivery. I have a few comments for your consideration listed below.

1.     During the thiol-ene ration, there are three active sites of the DHP-(AGE)3, as shown in figure 1b; I wonder which one the oligonucleotide prefers to bind first and how to control the binding sites. It is better to consider moving the part of figures S6 and S7 into the main manuscript to explain the situations clearly. 

2.     Flowing the Q1, will the binding sites of the DHP-(AGE) significantly affect the final product of the nucleolipid DHP-(AGE)3-oligonucleotide or the formulation of the SNAs? Do you consider the less or more binding sites, such as (AGE)2 or (AGE)4?

3.     Is this a model protocol to produce the nucleolipid DHP-(AGE)3-oligonucleotide? Have you considered other oligonucleotides excepted using in this manuscript? Are the kinds of oligonucleotides affect the synthesis procedure? Some discussion might be suitable for this point.

4.     I found the hydrodynamic diameter of the plain DPPC/Chol liposomes is larger than the SNAs (Table 1 and figure 4), which might be required more explanation and discussion. The additional oligonucleotide on the surface might induce steric hindrance, as shown in figure 3, which will increase the hydrodynamic size of the SNAs. What’s your assumption of the position of the oligonucleotide will be in the nanoparticles? Will it bend and insert into the lipid or be allied outside the surface of the liposome?

5.     How does the FITC co-assemble with SNAs? Will it be outside of the surface of the SNAs? Any DLS data for the SNAs cooperated with FITC? The DLS results of the SNAs-FITC nano complex could be used to compare with the SNAs themselves. I assume the SNAs-FITC nano complex will be larger than the SNAs.

6.     How does the laurdan co-assemble with SNAs? Will it be inside the SNAs or outside, similar to the FITC? If it is inside, the liposome leakage experiment might be helpful to identify the encapsulation of the laurdan, i.e. forming the mixture and adding Triton™ X-100 to trigger the leakage of the nanoparticle and release the dye. 

7.     Overall, much more attention should be taken to the section on the labelling of SNAs with laurdan and FITC since the different co-assembly behaviour will significantly affect the final fluorescent behaviour.

Author Response

Reviewer 2

Using a typical click chemistry thiol-ene reaction, the author synthesized the nucleolipid DHP-(AGE)3-oligonucleotide, formulated with phospholipid DPPC and cholesterol as SNAs for model dyes encapsulation. The synthesis protocol is well-planed, with the produced compound/nanoparticle systematically characterized and identified via various techniques, including NMR, HPLC, DLS, UV-Vis, Cryo-TEM, etc., potentially bringing further applications of the SNAs such as drug or gene delivery. I have a few comments for your consideration listed below.

  1. During the thiol-ene ration, there are three active sites of the DHP-(AGE)3, as shown in figure 1b; I wonder which one the oligonucleotide prefers to bind first and how to control the binding sites. It is better to consider moving the part of figures S6 and S7 into the main manuscript to explain the situations clearly. 

Response: This concern is related to the concern 2 of Reviewer #1. The oligonucleotide prefers to bind to the outermost binding site of the (AGE)n spacer due to steric considerations. The (number of) binding sites at which the conjugation reaction takes place is controlled by the ratio DHP-(AGE)n:thiolated species (AET.HCl, PEGMET, thiolated oligonucleotide) as demonstrated by the model reactions. We are afraid that moving figures from the model reactions to the main text would change the balance in the manuscript aiming at demonstrating synthesis of an original nucleolipid rather than investigation of a thiol-ene click reaction. Nevertheless, the text on lines 234-240 was modified to explain the situation more clearly.

  1. Flowing the Q1, will the binding sites of the DHP-(AGE) significantly affect the final product of the nucleolipid DHP-(AGE)3-oligonucleotide or the formulation of the SNAs? Do you consider the less or more binding sites, such as (AGE)2or (AGE)4?

Response: It is quite unlikely that changing the binding position in DHP-(AGE)3-oligonucleotide would affect the properties of the final product or the formulation of the vesicular SNAs since the spacer is very short. However, if the spacer was longer and more oligonucleotide strands were attached to it, changes in the properties could be anticipated – e.g., the final product could be rather hydrophilic and not able to intercalate in the bilayer membrane of the liposomes.

  1. Is this a model protocol to produce the nucleolipid DHP-(AGE)3-oligonucleotide? Have you considered other oligonucleotides excepted using in this manuscript? Are the kinds of oligonucleotides affect the synthesis procedure? Some discussion might be suitable for this point.

Response: Yes, the reaction/protocol is applicable to thiolated oligonucleotides independently from the type (DNA or RNA) and sequence (non-specific as in this study or specific, e.g., for gene silencing or other purpose). A sentence in the Conclusions section was added to emphasize this (lines 342-343).

  1. I found the hydrodynamic diameter of the plain DPPC/Chol liposomes is larger than the SNAs (Table 1 and figure 4), which might be required more explanation and discussion. The additional oligonucleotide on the surface might induce steric hindrance, as shown in figure 3, which will increase the hydrodynamic size of the SNAs. What’s your assumption of the position of the oligonucleotide will be in the nanoparticles? Will it bend and insert into the lipid or be allied outside the surface of the liposome?

Response: Here we note that the vesicular SNAs were prepared by co-assembly of DPPC, Chol, and nucleolipid, rather than by grafting on the surface of pre-formed liposomes. This is clearly stated in lines 114-125 and 260-265. The size decrease might be somewhat counterintuitive in first sight but can be explained in curving of the bilayer (ergo, formation of smaller in size vesicles) resulting from repulsive interactions between the oligonucleotide strands. This is discussed in lines 273-277. With regard to the comment for the position of the oligonucleotide, we note that the oligonucleotide strands are charged and highly hydrophilic and therefore highly unlikely to penetrate and insert in the hydrophobic domain of the phospholipid bilayer. They are pointed to the aqueous phases as schematically depicted in Figure 3 and adopt an unextended conformation as discussed in lines 307-310.

  1. How does the FITC co-assemble with SNAs? Will it be outside of the surface of the SNAs? Any DLS data for the SNAs cooperated with FITC? The DLS results of the SNAs-FITC nano complex could be used to compare with the SNAs themselves. I assume the SNAs-FITC nano complex will be larger than the SNAs.

Response: FITC does not co-assemble with SNAs; it covalently binds to DNA at the 5'-end (lines 315-316). It is supposed to be outside of the surface of the SNAs since the labelling was performed on the pre-formed vesicles. The size and zeta potential of the labelled vesicular SNAs practically did not change – the values were within the standard deviations (see Table 1) – possibly due to the relatively low molar mass and compact molecule of the fluorophore as well as to the unextended conformation of the oligonucleotide strands. The text in lines 318-322 was changed to acknowledge this finding.

  1. How does the laurdan co-assemble with SNAs? Will it be inside the SNAs or outside, similar to the FITC? If it is inside, the liposome leakage experiment might be helpful to identify the encapsulation of the laurdan, i.e. forming the mixture and adding Triton™ X-100 to trigger the leakage of the nanoparticle and release the dye. 

Response: Laurdan, 6-lauroyl-2-(N,N-dimethylamino)naphthalene, is a lipophilic dye, used to investigate membrane quality of phospholipid bilayers (lines 316-317). Adding Triton™ X-100 would undoubtedly convert the vesicles into micelles in which laurdan, being lipophilic/hydrophobic, would be solubilized rather than released. The leakage experiments, suggested by Reviewer 2, are more appropriate for hydrophilic dyes encapsulated in the internal aqueous pool of the vesicles.

  1. Overall, much more attention should be taken to the section on the labelling of SNAs with laurdan and FITC since the different co-assembly behaviour will significantly affect the final fluorescent behaviour.

Response: With this section we sought qualitative evidence for successful formation of vesicular SNA structures and their loading/labelling with either hydrophilic or lipophilic dyes. Leakage experiments, loading capacity, release experiments, fluorescent behavior are topics of research that is in progress.

Round 2

Reviewer 1 Report

The manuscript has been strengthened after the revision, and the authors have addressed all my previous concerns. I would recommend acceptance of the current version.

Reviewer 2 Report

The authors addressed all of my concerns. Thanks.